# Interventions to promote family member involvement in adult critical care settings: a systematic review

Andreas Xyrichis ![ORCID],[1] Simon Fletcher,[2] Julia Philippou,[1] Sally Brearley,[1] Marius Terblanche,[3] Anne Marie Rafferty[1]

¹Florence Nightingale Faculty of Nursing, Midwifery and Palliative Care, King's College London, London, UK
²Health, Social Care and Education, Kingston and St Georges University London, London, England, UK
³Critical Care, Guy's and St Thomas' NHS Foundation Trust, London, UK

**Correspondence to**
Andreas Xyrichis;
andreas.xyrichis@kcl.ac.uk

## ABSTRACT

**Objective** To identify, appraise and synthesise evidence of interventions designed to promote family member involvement in adult critical care units; and to develop a working typology of interventions for use by health professionals and family members.

**Design** Mixed-method systematic review.

**Data sources** Bibliographic databases were searched without date restriction up to June 2019: MEDLINE, EMBASE and CINAHL; the Cochrane Central Register of Controlled Trials, Joanna Briggs and Cochrane Libraries. Back issues of leading critical care and patient experience journals were manually searched, as were the reference lists of included studies. All evaluation studies of relevant intervention activities were included; all research designs and outcome measures were eligible. Due to heterogeneity in interventions, designs and outcome measures, the synthesis followed a narrative approach. Service users met with the research team termly.

**Results** Out of 4962 possible citations, a total of 20 studies were included. The overall evidence base was assessed as moderate to weak. Six categories of interventions were identified: environmental unit changes (n=2), web-based support (n=4), discussion-based support (n=6), multicomponent support (n=4), participation in rounds (n=3) and participation in physical care (n=1). Clinical and methodological heterogeneity across studies hindered meta-analysis, hence a narrative synthesis was pursued. Six main outcomes were identified, grouped under two categories: (i) involvement outcomes: communication (*mean difference* ranged from 6.39 to 8.83), decision-making (*mean difference* ranged from −0.8 to 5.85), satisfaction (*mean difference* ranged from 0.15 to 2.48); and (ii) health outcomes: family trauma (*mean difference* ranged from −7.12 to 0.9), family well-being (*mean difference* ranged from −0.7 to −4), patient outcomes (*relative risk* ranged from 1.27 to 4.91). The findings from the qualitative studies were thematically analysed to identify features of the interventions that participants perceived to influence effectiveness. Synthesised into five overarching categories (practicality, development, interaction, reflexivity and bridging), these can serve as principles to inform the future design and development of more refined family member involvement interventions.

**Conclusions** Future interventions should be developed with much closer family member input and designed by considering the key features we identified. We call for

### Strengths and limitations of this study

► We completed a comprehensive mixed-method systematic review of available evidence on interventions for family member involvement in adult critical care.
► We involved a service users and carers group throughout the systematic review process, which included 16 hours of in-depth group discussions.
► Study screening, selection, quality assessment and data extraction were completed independently and in duplicate by two review authors.
► Methodological and clinical heterogeneity across studies prevented us from pursuing a meta-analysis.
► Qualitative evidence was synthesised thematically to identify features of the interventions that participants perceived to influence effectiveness.

future interventions to be multilayered and allow for a greater or lesser level, and different kinds, of involvement for family members. Choice of intervention should be informed by a baseline diagnostic of family members' needs, readiness and preparedness for involvement.

**PROSPERO registration** CRD42018086325.

## BACKGROUND

Family member involvement in healthcare is recognised by policy-makers and the public as an invaluable aspect of healthcare provision. The move towards the more structured integration of family members is consistent with the call for greater patient-centred care, long invoked by a (2001) report from the Institute of Medicine.[1] Emphasising the potential benefits in actively involving family members in decision-making processes and care management was regarded as a key contributor to the reduction of the risk of medical error and a broader improvement in quality. The idea that a 'fully-engaged' public[2] should be encouraged has also been evident in UK contexts; however, governmental scrutiny over the proceeding decade revealed that this has yet to be comprehensively realised. Two strategic reports from NHS England[3 4]

explored a practical move towards a healthcare system which is built around the patient, incorporating families representing a key means of achieving this.

For the purposes of the present review, family member involvement refers to the activities of different professionals to improve care by ensuring family members are involved in decision-making, sharing of information, power and responsibility for patient needs and choices. Family member involvement is especially relevant in the intensive care unit (ICU), where it can have profound consequences for care decisions, delivery and outcomes; this is partly because ICU patients are rarely in a position to communicate directly with clinicians and recollect their ICU experience, which means that the responsibility for this often lies with their family members. While this is a concern in healthcare systems worldwide, and especially in the UK, a recent mixed-methods study found there is still great variation in family satisfaction across ICUs in England, Wales and Northern Ireland.[5]

Studies over the past decade have confirmed that family members can have a positive influence on a patients' care and recovery from ICU; but also that family members themselves can be affected even after discharge.[6–11] In particular, within the first few days after ICU admission, family members can show signs of anxiety, depression and stress; report difficulties in understanding the information clinicians try to communicate with them, and those who suffered a bereavement are at risk of generalised anxiety, panic attacks, depression and post-traumatic stress disorder (PTSD).[7] For example, one study[6] surveyed family members of patients who have been in ICU to find that 90 days after discharge more than a third (34%) of them suffered from PTSD symptoms. In addition, they noted higher rates (48%) among those family members who indicated the information they were given was incomplete.

While there is a growing evidence base on patients, family members and clinicians' perceptions of involvement in ICU care, we are still missing a standardised, evidence-based approach to facilitate this process. Indeed, a recent scoping review[10] sought to investigate the extent and range of literature on this topic, finding evidence of a growth in papers with over 100 reports identified. However, they did not seek to assess the quality of the evidence nor did they examine the type of interventions available, and key features that foster effectiveness. While this is an area of growing interest, we are still unclear about the range and quality of interventions available to promote family member involvement in ICUs.

## METHODS

Our review sought to answer the following question: What are the available interventions, and which are most effective, for fostering family member involvement in adult critical care settings? Four objectives were set: (a) undertake a comprehensive and systematic search of published and unpublished studies reporting on interventions that promote involvement in adult critical care; (b) robustly assess the quality of empirical evidence for all included studies; (c) generate a detailed description and synthesis of interventions and their associated outcomes; and (d) classify interventions and outcomes in order to develop a typology of interventions, outlining key factors that support or impede involvement.

### Study eligibility criteria

We included evaluation studies of any design, including experimental and quasiexperimental studies, as well as action research, case study and ethnographic designs. We considered reports of any kind of interventions as long as they were intended to promote the participation of family members in adult critical care. We included studies that reported on a mixture of relevant outcomes, including standard measures such as the Hospital Anxiety and Depression Score (HADS); and non-standard but important indicators such as family satisfaction. We excluded non-evaluation papers and those with populations, outcomes and settings that did not match our brief. One study reported on a bundled intervention including awakening and breathing coordination, delirium monitoring and management, early mobility and family-centred care; this study had to be excluded because we could not discern outcomes specifically linked to the family involvement component of the bundle.

### Search for evidence

We searched (June 2019) MEDLINE, EMBASE, CINAHL and the Cochrane Central Register of Controlled Trials following a systematic approach without date restriction. We also performed searches for clinical studies through the WHO Trials Registry. We hand searched recent back

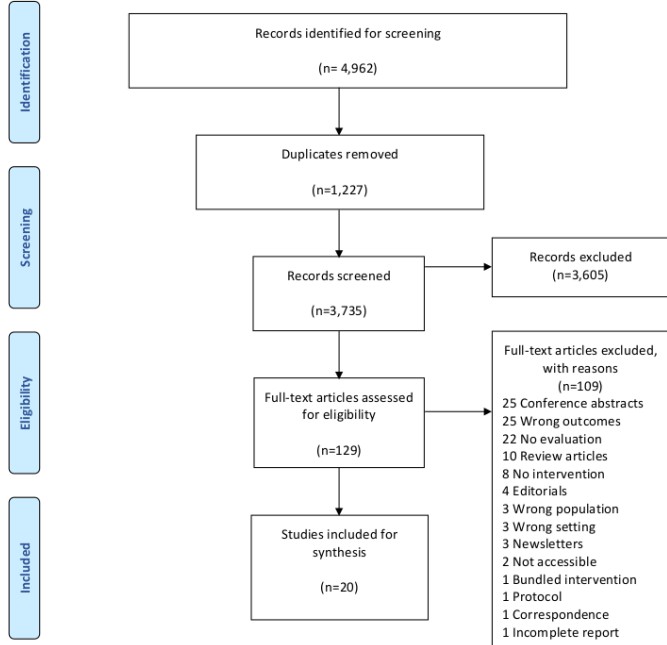

**Figure 1** Flow diagram.

**Table 1** Included studies

| Author | Year | Country and setting | Design | Sample size | Research approach | Intervention |
|---|---|---|---|---|---|---|
| Allen et al[32] | 2017 | USA—surgical ICU | Non-randomised, before and after study | Patients: n=847 Family members: n=429 | Quantitative | Engaging family members on rounds |
| Almoosa et al[22] | 2009 | USA—medical ICU | Prospective, two-centre observational study | Patients: n=85 Family members: n=85 | Quantitative | Cariopulmonary Resuscitation (CPR) discussions |
| Choi and Bosch[16] | 2013 | USA—neuro ICU and trauma ICU | Comparative observational study | Patients: n=81 Family members: n=81 | Quantitative | Patient and family-centred unit design |
| Cray[28] | 1989 | USA—medical ICU | Post hoc evaluation study | Patients: n=76 Family members: n=76 | Quantitative | Family intervention programme |
| Dalal et al[18] | 2015 | USA—medical ICU (oncology unit) | Post hoc evaluation study | Patients: n=26 Family members: n=77 | Quantitative | Patient-centred toolkit |
| Davidson et al[29] | 2010 | USA—mixed ICU | Feasibility study | Patients: n=30 Family members: n=22 | Quantitative | Family support programme, based on facilitated sensemaking |
| Dykes et al[19] | 2017 | USA—medical ICU | Non-randomised, before and after study | Patients: n=58 Family members: n=156 | Quantitative | Patient engagement communication and technology (PROSPECT) |
| Ernecoff et al[20] | 2016 | USA—medical ICU | Qualitative interview study | Family members: n=30 Staff members: n=28 | Qualitative | Tablet-based support tool |
| Garrouste-Orgeas et al[23] | 2016 | France—surgical ICU | Randomised-controlled trial with nested qualitative study | Patients: n=100 Family members: n=88 | Mixed-methods | Proactive participation of a nurse in family conferences |
| Hollman Frisman et al[24] | 2018 | Sweden—ICU | Qualitative interview study | Patients: n=8 Family members: n=10 | Qualitative | Health-promoting conversations |
| Huang et al[25] | 2018 | USA—neuroscience ICU | Prospective, single-centred observational study | Family members: n=263 | Quantitative | Primary care physician involvement in decision-making in the ICU |
| Huffines et al[21] | 2013 | USA—surgical ICU | Non-randomised, before and after study | Family members: n=48 | Quantitative | Family supportive care algorithm |
| Marshall et al[30] | 2016 | Australia—general ICU | Feasibility study | Family members: n=51 | Qualitative | Multifaceted family-centred nutrition intervention |

**Table 1** Continued

| Author | Year | Country and setting | Design | Sample size | Research approach | Intervention |
|---|---|---|---|---|---|---|
| Jacobowski et al[33] | 2010 | USA—medical ICU | Non-randomised, before and after study | Family members: n=111 | Quantitative | Family rounds |
| Prichard and Newcomb[35] | 2015 | USA—trauma ICU | Quasi-experimental pilot study | Family members: n=30 | Quantitative | Hand massage |
| Randall-Curtis et al[26] | 2016 | USA—general ICU | Randomised-controlled trial | Patients: n=168 Family members: n=268 | Quantitative | Communication facilitator |
| Rippin et al[17] | 2015 | USA—neuroscience ICU | Comparative observational study | Family members: n=54 Staff members: n=18 | Qualitative | Family-centred unit design |
| Shaw et al[27] | 2014 | USA—general ICU | Non-randomised, before and after study | Patients: n=121 Family members: n=121 | Quantitative | Multidisciplinary team training to enhance family communication in the ICU |
| Weber et al[34] | 2018 | USA—neuroscience ICU | Non-randomised, before and after implementation study | Family members: n=141 | Quantitative | Family rounds |
| White et al[31] | 2018 | USA—two neuro ICU, two mixed ICU, one medical ICU | Multicentre, stepped-wedge, cluster-randomised | Patients: n=1420 Family members: n=1106 | Quantitative | Multicomponent family-support intervention |

CPR, Cardiopulmonary resuscitation; ICU, intensive care unit; PROSPECT, Promoting Respect and Ongoing Safety through Patient Engagement Communication and Technology .

issues of key critical care and patient experience e-journals, scanned reference lists of identified reviews and included articles; searched our personal libraries and consulted with experts in the field. GoogleScholar was also searched. Non-commercially published (Grey) literature was sought through the OpenGrey and the GreyLit Report databases. A standardised search strategy was developed with the input of an information specialist and applied to all databases, utilising both MeSH and free-text terms (online supplemental file).

## Data management, screening, extraction and quality assessment
Search results were imported into Covidence, the standard production platform for Cochrane Reviews. Two reviewers (AX and SF) independently screened all citations in duplicate against the inclusion criteria. Any disagreement was resolved through discussion or by involving a third reviewer (JP). Reasons for excluding papers read in full were recorded and a preferred reporting items for systematic reviews and meta-analyses

flowchart was completed. Standardised data extraction forms were used by two reviewers (SF and AX) to independently extract key information. Any disagreements found at this stage were also resolved through a consensus approach, or the involvement of a third reviewer (JP). We assessed each included paper for methodological rigour using an established rating approach utilised successfully in a number of previous mixed-method systematic reviews.[12 13] This rating approach assesses 'quality of study' to consider: appropriate fit between study design/research questions and use of appropriate analyses; and, 'quality of information' to assess for a clear rationale for the intervention, good contextual information and risk of bias. Quality assessment was undertaken independently by two reviewers (AX and SF) in duplicate. For each quality domain, we described the procedures reported in the paper, using verbatim quotes as necessary.

## Data synthesis
The overall approach to the synthesis followed a segregated methodologies approach.[14] Accordingly, a separate

synthesis of quantitative and qualitative data was initially completed followed by a mixed-method synthesis. Quantitative studies were grouped depending on study design, and results summarised using descriptive statistics. Because heterogeneity precluded statistical combination through a meta-analysis, a narrative approach to the quantitative synthesis was pursued. Synthesis of qualitative data followed the best-fit framework approach, involving identification and grouping of thematic categories around factors influencing effectiveness of family member involvement interventions.[15] The mixed-method synthesis utilised the quantitative data to inform on the measured effects, while the qualitative data informed on the perceived effects. In this way, the qualitative and quantitative data acted as complementary rather than as confirmatory of each other. A key element of the mixed-method synthesis was to classify interventions and outcomes in order to develop a typology of family involvement interventions in ICU. To develop the typology, we identified categories for the different interventions used in the studies by noting their distinguishing characteristics. We also formed and refined key family involvement features of the interventions, such as required time commitment from families and staff, opportunities and challenges, and cost. We tabulated these alongside an indication of the quality level of the current body of evidence, and the expected impact on important engagement and clinical outcomes. The typology was developed in collaboration with a service users and carers group over two face-to-face meetings, during which the overall findings of the review and ways of representing these visually were discussed. The resulting typology is illustrated in a matrix-like format presenting the intervention categories in a continuum from low to high involvement.

## Patient and public involvement

From the outset, the development of this project was grounded in patient and public involvement (PPI) since family member involvement in ICU has been identified as a priority area for research by the James Lind Alliance (http://www.jla.nihr.ac.uk) through a priority setting partnership involving patients, carers and clinicians. ICU clinicians and patients raised our awareness on the lack of evidence-based advice on interventions that promote family involvement in ICU, and the challenges this introduces to daily practice. We included a PPI expert on our team (SB) and a frontline ICU clinician (MT) with whom we developed the project application and review protocol. We also invited feedback on our initial plans from the UK ICUSteps Charity ensuring that the project was attuned to the sensitivities and concerns of service users. As part of the review process, a service users and carers group was established, the members of which were recruited through promotion in social media, the UK ICUSteps charity and the National Institute for Health Research INVOLVE website. Consisting of eight ICU survivors and carers, the group was consulted through each step of the process and engaged in four strategically positioned meetings leading to over 16 hours of focused and in-depth face-to-face discussions concerning the design, conduct, reporting and dissemination of the review. By having an engaged service user and carer group work with us throughout the completion of this review, we were able to look beyond the evidence by exploring the experiences and views of service users about the different interventions identified.

## RESULTS
### Overview of included studies
#### Search results

Out of 4962 possible citations, a total of 20 studies were included (figure 1). Published between 1989 and 2018, they were predominantly from North American contexts (n=17), and from a range of ICU settings including surgical, neuroscience, medical oncology and trauma ICUs. One study came from a French setting, and others from Swedish and Australian ICU settings.

#### Study designs

Most of the studies followed a quantitative approach (n=15), but we also included four qualitative and one mixed-methods study (see table 1). Specifically, quantitative studies included randomised-controlled trials, cluster-randomised and stepped-wedge trials, non-randomised before and after studies and prospective observational multicentre studies. Qualitative data from all studies came from semistructured interviews. Sample sizes across the quantitative studies ranged from 30 to 1106 family members; and in the qualitative studies from 10 to 54 family members.

#### Intervention types

Interventions were grouped based on their distinguishing characteristics (summarised in table 2), and positioned within a continuum of low to high involvement: (i) environmental unit changes (n=2),[16 17] (ii) web-based support (n=4),[18–21] (iii) discussion-based support (n=6),[22–27] (iv) multicomponent support (n=4),[28–31] (v) participation in rounds (n=3)[32–34] and (vi) participation in physical care (n=1).[35] Environmental unit change interventions consisted of complete structural redesign of an existing ICU with the specific objective being enhanced family member presence. Web-based and electronic interventions were approaches which utilised information and communication technology to facilitate information sharing and asynchronous communication. Discussion-based interventions generally consisted of one off or repeated face-to-face conversations which took place in the unit between family members and healthcare professionals. Multicomponent interventions comprised more than one technique to engage, educate and communicate with family members in the ICU. Family involvement in rounds interventions enabled the family members to attend and watch the daily rounds process, with opportunity to raise questions and clarify issues. Physical participation in care consisted of physical touch between

**Table 2** Intervention characteristics

| Author | Activity type | Intervention | Purpose | Procedure | Participants | Beneficiary | Outcomes |
|---|---|---|---|---|---|---|---|
| Allen et al 2017[32] | Rounds based | Family member involvement in rounds | Engaging and integrating ICU patients' family members and surrogates on daily rounds | On admission to the ICU, a family member was invited to participate on daily rounds with the critical care team | Family members, nurses, physicians | Family members, nurses, physicians | ▲ Significantly increased family member knowledge<br>▲ Strengthened relationship between family members and doctors<br>▲ Nurses reported greater work enjoyment<br>▲ Improved communication<br>▲ Reduction in workload<br>▲ Increased support for patient and family-centred care<br>▲ Increased physician satisfaction post intervention |
| Almoosa et al 2009[22] | Discussion based | Cardiopulmonary Resuscitation (CPR) discussions | The implementation of structured, informed CPR specific conversations with family members and surrogates from relevant physicians | Physician initiated CPR conversation about: chest compressions, electrical cardioversion, mechanical ventilation | Physicians, residents, fellows or family | Family members | ▲ Satisfaction with CPR discussions was higher after the intervention |
| Choi and Bosch 2013[16] | Unit design | Patient and family-centred unit design | Unit redesign undertaken to enable physical space to more broadly facilitate family interaction with patient, and family interaction with staff | The intervention unit included larger spaces and more comfortable accommodations for visitors | Family members | Family members and patients | ▲ Increased family interactions with patient<br>▲ Increased family interactions with staff |
| Cray et al 1989[28] | Multicomponent | Family intervention programme | Multicomponent intervention to enhance family involvement in, and understanding of, their loved one's condition in ICU | Intervention components included: family conference, telephone conversation, visit by nurse, follow-up visit to the intermediary care unit | Family members, clinical nurse specialists | Family members | ▲ Strong agreement that the intervention helped family members to understand their loved one's illness and benefit other families<br>▲ High satisfaction with intervention components |
| Dalal et al 2016[18] | Web based/ electronic | Patient-centred toolkit (PCTK) | The PCTK was a suite of web-based patient-facing and provider-facing tools designed to facilitate collaborative decision-making by providing access to tailored educational content and facilitating patient-provider communication | The PCTK was accessible by patients, caregivers and providers from any web-enabled device connected to the hospital's secure intranet | Patients and caregivers | Patients and family members | ▲ The majority of patients and caregivers surveyed were satisfied or extremely satisfied with the intervention |
| Davidson et al 2010[29] | Multicomponent | Family-support programme, based on facilitated sensemaking | A systems-based patient-centred care and engagement programme | The intervention consisted of two main components: personalised instruction and provision of family visiting kits | Family members, Clinical nurse specialists | Family members | ▲ High rate of instrument reliability<br>▲ A breakdown of the top 10 needs of family members<br>▲ A ranking of interventions' helpfulness<br>▲ Identification of any additional needs |

Continued

**Table 2** Continued

| Author | Activity type | Intervention | Purpose | Procedure | Participants | Beneficiary | Outcomes |
|---|---|---|---|---|---|---|---|
| Dykes et al 2017[19] | Web based/electronic | Promoting respect and ongoing safety through patient engagement communication and technology (PROSPECT) | A systems-based patient-centred care and engagement programme designed to enhance patient and care partner experience in the ICU | 60-min training session to facilitate patient-centred care in healthcare professionals. A web-based toolkit to apply the training, delivery of the patient satisfaction model from nurses to patients in ICU | Nurses and patients | Nurses and patients | ▲ Relative reduction in the rate of adverse effects ▲ Improvements in patient satisfaction ▲ Improvements in care partner satisfaction |
| Ernecoff et al[20] 2016 | Web based/electronic | Tablet-based and video-driven communication and decision support tool | A tool was conceptualised to: (1) prepare the family for conversations with clinicians, (2) give clinicians tailored information about the family and patient in advance of the family meeting, (3) promote a personalised relationship between clinician and family, and (4) provide general decision support to surrogates | The sections of the tool for surrogates were: (1) orienting surrogates to the ICU, (2) explaining principles of surrogate decision-making, (3) providing a question prompt list and opportunity to write down questions, (4) a values clarification exercise, (5) education about treatment pathways (eg, life-prolonging treatment, comfort-focused treatment and a time-limited trial of ICU care), (6) eliciting surrogates' prognostic information, and (7) providing psychosocial resources | Family members (surrogates) | Family members | ▲ Enhancing and supplementing communication between surrogates and the clinical team ▲ Leveraging surrogates' downtime before and between clinician-family meetings ▲ Helping surrogates to consider the patient's values and treatment options ▲ Allowing for repetition and review of information |
| Garrouste-Orgeas et al 2016[23] | Discussion based | Proactive participation of a nurse in family conferences | Integration of nursing staff in family conferences focused around: naming emotions, expressing understanding, showing respect, articulating support for the patient and exploring the family's emotional state | All family members who wished to attend were escorted to the room. Briefly, an open question was asked first to encourage the family members to express themselves. A substantial proportion of the time was devoted to listening to the family, and the professionals used simple words to enhance comprehension. At the end of the conference, the family was allowed enough time to ask all the questions they had | Physicians, nurses, family members | Family members | ▲ Family members reported that the conferences allowed them to receive and assimilate information ▲ Be listened to regarding both their positive and negative feelings ▲ Receive compassion and respect |

**Table 2** Continued

| Author | Activity type | Intervention | Purpose | Procedure | Participants | Beneficiary | Outcomes |
|---|---|---|---|---|---|---|---|
| Hollman Frisman et al 2018[24] | Discussion based | (Nurse led) Health-promoting conversations with families | The aim of the conversations was to create a context for change related to the families' problems and resources | Discussion of the aim of the conversation series and the families' and nurses' expectations about the conversations and each other's roles. The three conversation sessions focused on topics that the families considered important, and the dialogue and questioning intended to identify resources within and outside the family. At the end of each conversation, the nurse offered a short reflection on how the family members had experienced the session. A closing letter was sent to the family 2–3 weeks after the last conversation to summarise further possibilities for reflection | Nurses and family members | Family members | ▲ Health-promoting conversations led to increasing emotional openness<br>▲ Enhanced consciousness regarding illness<br>▲ Greater family member satisfaction<br>▲ A valuable sense of confirmation<br>▲ General promotion of family well-being |
| Huang et al 2018[25] | Discussion based | Primary care physician (PCP) involvement in decision-making in the ICU | The study evaluated survey results which measured family member satisfaction with general ICU care and shared decision-making between primary care physicians and family members | Examined involvement of the patient's primary care physician and shared decision-making | Primary care physicians | Family members | ▲ A higher proportion reporting PCP involvement felt completely satisfied with their inclusion in the ICU decision-making process |
| Huffines et al 2013[21] | Web based/electronic | Family supportive care algorithm | The goal of the interventions in the 24-hour bundle was to inform families about the importance of their participation in decision-making and to inform them of the resources available to help them participate in decision-making | Within 24 hours of admission, a member of the intensivist team met with the family. Also within the first 24 hours after a patient's SICU admission, the patient's family was encouraged to watch an on-demand 10 min video.<br>72 hours:<br>Reaffirm and ensure family is supported. Encouraged to participate in decision-making in rounds.<br>96 hours:<br>Implement a family meeting with the multidisciplinary team. | Bedside nurses | Family members | ▲ Mixed to positive outcomes when measuring family satisfaction<br>▲ Staff teamwork<br>▲ Participation in decision-making<br>▲ Frequency of support |

Continued

**Table 2** Continued

| Author | Activity type | Intervention | Purpose | Procedure | Participants | Beneficiary | Outcomes |
|---|---|---|---|---|---|---|---|
| Marshall et al 2016[30] | Multicomponent | Multifaceted, family-centred nutrition intervention | To assist families in discussing nutritional goals with health professionals | Consisted of: family interviews, focused education sessions, and nutrition diaries | Dieticians, family nurses | Families and patients | ▲ Perceptions around how nutrition education can be improved<br>▲ The diversity of experience relating to the provision of in-hospital nutrition therapy<br>▲ Continuity of existing family member involvement approaches<br>▲ The importance of families as advocates |
| Jacobowski et al 2010[33] | Rounds based | Family rounds | To enhance communication between medical and nursing staff and the families of ICU patients | The attending physician provided a summary for the family using understandable, lay language and the family was offered an opportunity to ask questions of the team | Physicians, nurses, family members | Physicians, nurses, family members | ▲ Communication regarding condition improved significantly as did decision-making support<br>▲ There was a decline in the number of family members who thought that they had sufficient time to address their questions and concerns |
| Prichard and Newcomb 2015[35] | Physical | Hand massage | To provide physical relief to patients in the ICU through hand massage | Participants were taught to administer hand massage in compliance with the M technique, a registered method of simple, structured touch that has been used on critically ill patients with positive effects. For this study, the technique was used on hands. Participants applied the intervention twice daily for 5 min per session for 3 consecutive days | Family members, patients | Family members, patients | ▶ Anxiety was greatly reduced in the treatment group |
| Randall-Curtis et al 2016[26] | Discussion based | Communication-facilitator | To understand the family's concerns, needs and communication characteristics | Consisted of facilitated: interviews, meetings, communication and emotional support; as well as facilitator participation in family conferences and 24-hour follow-up | Nurse or social worker trained to improve communication between the ICU team and the family by acting as a communication facilitator or navigator | Family members | ▶ Adjusted depression scores were lowered alongside ICU costs |

**Table 2** Continued

| Author | Activity type | Intervention | Purpose | Procedure | Participants | Beneficiary | Outcomes |
|---|---|---|---|---|---|---|---|
| Rippin et al 2015[17] | Unit design | Family-centred unit (FCU) design | Unit designed to prioritise and engage the family in the ICU context | The FCU physically integrated family into the fabric of the unit. Nurses worked between centralised nursing stations and decentralised alcoves just outside patient rooms for improved monitoring and safety. Rooms were larger (245 sq ft) with more space around the bedside | Nurses and family members | Family members | ▲ Reduction in problematic bedside copresence (or overcrowding)<br>▲ A continuity of preintervention distribution patterns in which nurse and family remained clustered in their respective 'domains'<br>▲ Nurse perspectives which acknowledged the unpredictability and complexity of the ICU and the subsequent mixed results which this newly implemented unit design encouraged |
| Shaw et al 2014[27] | Discussion based | Multidisciplinary team (MDT) training to enhance family communication in the ICU | The training material was selected to improve communication. The training was also designed to foster team building and improve collaborative relationships among clinicians from multiple disciplines | Training included how to conduct and participate in patient/family conferences, addressing goals of care and giving critical information in the intensive care setting so that all caregivers might speak to patients and their families with a common voice. The training was designed to address known drivers of family satisfaction, as well as to address each of the 21 items being measured in the staff confidence survey | ICU team members | MDT team members and family members | ▲ Staff confidence and family satisfaction were shown to significantly improve |
| Weber et al 2018[34] | Rounds based | Family rounds | To improve family satisfaction in ICU experience by integrating them into rounds processes | After each session, the nursing leader recorded (1) whether family rounds occurred (2), how many families participated, and (3) how many patients the ICU team had rounded on that morning | Nursing staff | Family members | ▲ Family reported improved satisfaction with decision-making, frequency of communication, receiving emotional support and with coordination of care<br>▲ Percentage improvements in satisfaction scores were not large enough to reach statistical significance |
| White et al 2018[31] | Multicomponent | Multicomponent family-support intervention | Collaborative intervention between nursing staff and family members designed to support caregivers in the ICU | Nurses received advanced communication training. A family-support pathway was instituted. Intensive support for implementation was provided to each ICU by a quality-improvement specialist to incorporate the family-support pathway into clinicians' workflow | Nursing staff, ICU team, family members | Family members and nursing staff | ▲ Significant improvements were noted with family satisfaction regarding:<br>▲ the quality of communication with clinicians, and<br>▲ perceptions of patient centeredness<br>▲ The intervention led to an average reduction in the length of stay in the ICU of 3 days |

CPR, Cardiopulmonary resuscitation; ICU, intensive care unit; PCP, Primary care physician.

family members and patients being utilised for reciprocal benefit.

## Quality of evidence

The overall evidence base was assessed to be moderate to weak, with only a few exceptions of high-quality quantitative and qualitative studies (tables 3 and 4). The main weaknesses of the quantitative studies were related to inadequate randomisation and blinding, and lacking a control group for comparison purposes. The qualitative studies were in the main of a better quality, with key weaknesses being related to inadequate researcher reflexivity and theorisation of findings.

## Quantitative results

Six main outcome categories were identified from the studies analysed for the review, grouped under two groups. Involvement outcomes: communication, decision-making, satisfaction; and health outcomes: family well-being, family trauma, patient outcomes. A summary of key results is provided below, with detailed reporting of outcome measures available in the tables; in online supplemental tables 1–6 the outcome measures are presented according to intervention type.

## Involvement outcomes

Communication (table 5): improved communication was considered as an outcome in five studies,[27 31–34] the majority of which involved family participation in rounds, reporting on four measurement tools. Three studies[27 33 34] utilised the Family Satisfaction ICU (FS ICU 24) tool, specifically reporting on three dimensions: honesty of information, frequency of doctor communication and frequency of nurse communication. One study[31] reported on the Quality of Communication tool and another used a self-developed questionnaire.[32] Mean improvements were found in all studies, although not all results reached statistical significance. The biggest effect was seen on the frequency of doctor communication, with one study of family participation in rounds showing a statistically significant improvement of 60% (relative risk, RR: 1.60, CI: 1.18 to 2.17, p=0.004).[33]

Decision-making (table 6): six of the included studies[21 22 25 27 33 34] looked at decision-making as an outcome, most of which were discussion-based interventions, through six measurement tools. The 'decision making' subscale of the FS ICU 24 revealed improvements in three studies although only one,[27] targeting the whole interprofessional team, found a statistically significant result (MD: 5.85, p=0.05). Inclusion in decision-making was also considered, as the percentage change in family members reporting complete satisfaction or giving the highest score, with one study[25] which involved the patient's primary physician showing a statistically significant improvement of 23% (RR: 1.23, CI: 1.03 to 1.49, p=0.05). Studies also noted improvements with regard to 'control over patient care' (RR: 1.31, CI: 1.07 to 1.61, p=0.02),[25] 'support in decision-making' (RR: 1.39, CI:

1.09 to 1.79)[21] and 'share in decisions about care planning' (23%, RR: 1.52, CI: 1.03 to 2.24, p=0.009).[21]

Satisfaction (table 7): satisfaction was considered as an outcome in nine studies[18 19 21 25 27 28 31 33 34] using six different measures, and reporting on a mixture of discussion-based, rounds-based, multicomponent-based and web-based interventions. The majority of studies (n=4) measured family satisfaction with care using the FS-ICU, while the remaining studies used self-developed tools. One of the studies[19] also used the HCAHPS (Hospital Consumer Assessment of Healthcare Providers and Systems) Survey. This is a national, standardised, publicly reported survey of patients' perspectives of hospital care in the USA. Mean improvements on the FS ICU were found in all studies, ranging from 1.49 to 5.7, although only one[19] which used a web-based engagement intervention reached statistical significance (MD: 5.7, CI: 2.31 to 9.09, p<0.05). Statistically significant improvements were also noted on the HCAHPS (RR: 1.33, CI: 1.10 to 1.55, p<0.05).[19]

## Health outcomes

Family well-being was considered through measurements for anxiety and depression.

Anxiety (table 8): four studies, which encouraged family participation in discussions and personal care, considered anxiety as an outcome[23 26 31 35] using three different ways for measurement. Three studies[23 31 35] used the HADS. The fourth study[26] used the Generalised Anxiety Disorder Assessment (GAD-7). All four studies identified a reduction in family members' anxiety scores post the intervention, with the effect ranging from −0.34 to −4. The minimal clinically important difference of HADS has been suggested to be between −1.5 and −2; therefore, the reduction noted in these papers could be of clinical significance. However, only two of these studies found statistical significance.[23 35]

Depression (table 9): depression was an outcome also reported by three of the above studies[23 26 35] using two measures. Two studies[23 35] used the depression subscale of the HADS, while another[26] used the Patient Health Questionnaire (PHQ-9). Both studies that used the HADS identified a reduction in depression among family members, although only one[23] reached statistical significance (p=0.04) with a median reduction of 3.5. Results on the PHQ-9 also showed a statistically significant mean reduction in scores at 6 months (MD: −2.3, CI: −4.30 to −0.42, p=0.01).[26]

Family trauma (table 10): three studies considered the impact of interventions (multicomponent-based, web-based and discussion-based) on family member ICU-induced trauma using four different measurement tools:[23 26 31] PTSD Checklist, Impact of Event Scale (IES) and Peritraumatic Dissociative Experience Questionnaire. All three studies identified a trend showing a beneficial effect of family involvement interventions on severity of trauma through reduced severity scores, ranging from a reduction of −3 to −7 points on the various scales; however, none of these results reached statistical significance.

**Table 3** Quality assessment—quantitative studies

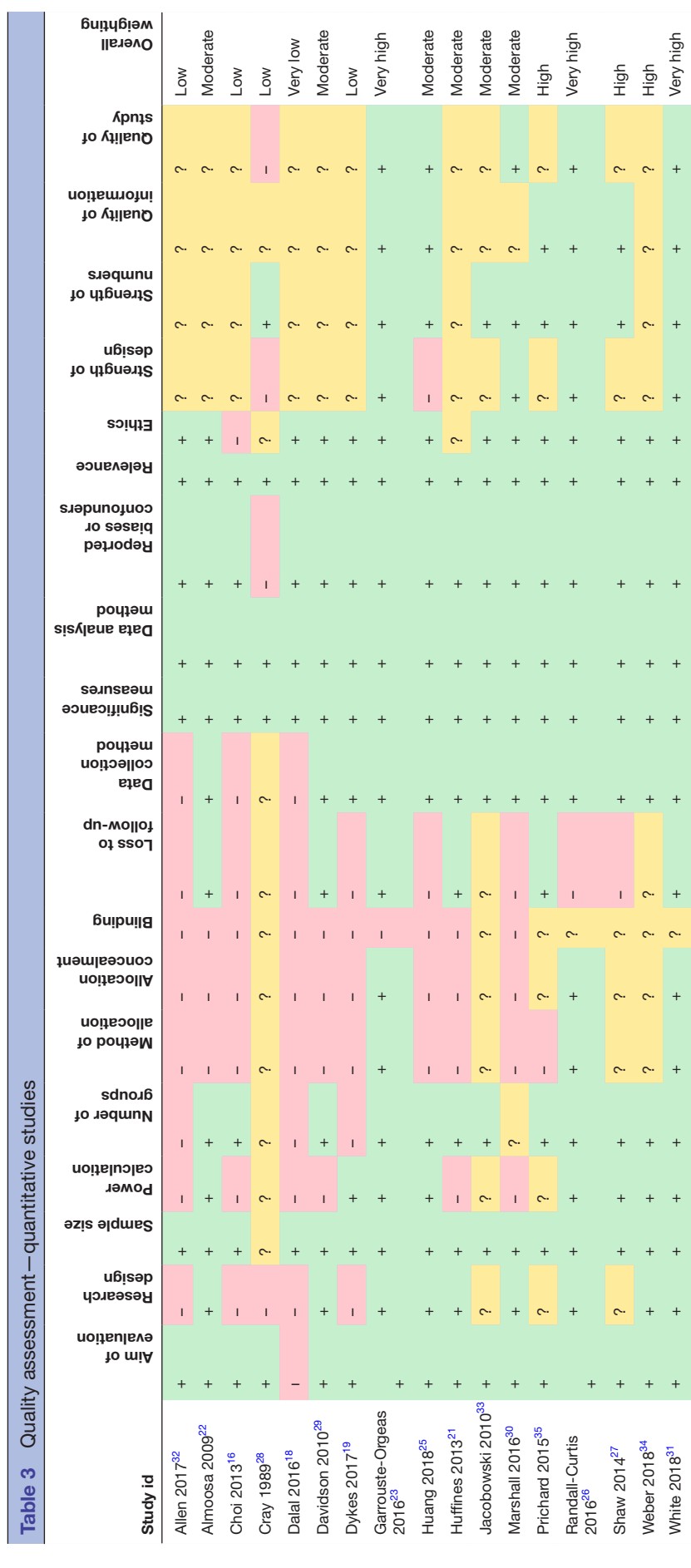

| Study id | Aim of evaluation | Research design | Sample size | Power calculation | Number of groups | Method of allocation | Allocation concealment | Blinding | Loss to follow-up | Data collection method | Significance measures | Data analysis method | Reported biases or confounders | Relevance | Ethics | Strength of design | Strength of numbers | Quality of information | Quality of study | Overall weighting |
|---|---|---|---|---|---|---|---|---|---|---|---|---|---|---|---|---|---|---|---|---|
| Allen 2017[32] | + | − | + | − | − | − | − | − | − | − | + | + | + | + | + | ? | ? | ? | ? | Low |
| Almoosa 2009[22] | + | + | + | + | + | − | − | + | + | + | + | + | + | + | + | ? | ? | ? | ? | Moderate |
| Choi 2013[16] | + | − | + | − | + | − | − | − | − | + | + | + | + | + | − | ? | ? | ? | ? | Low |
| Cray 1989[28] | + | − | ? | ? | ? | ? | ? | ? | ? | − | + | + | − | + | ? | − | + | ? | − | Low |
| Dalal 2016[18] | − | − | + | − | − | − | − | − | − | + | + | + | + | + | + | ? | ? | ? | ? | Very low |
| Davidson 2010[29] | + | + | + | − | + | − | − | + | + | + | + | + | + | + | + | ? | ? | ? | ? | Moderate |
| Dykes 2017[19] | + | − | + | + | + | − | − | − | − | + | + | + | + | + | + | ? | ? | ? | ? | Low |
| Garrouste-Orgeas 2016[23] | + | + | + | + | + | + | + | + | + | + | + | + | + | + | + | + | + | + | + | Very high |
| Huang 2018[25] | + | + | + | + | + | − | − | − | − | + | + | + | + | + | + | − | + | + | + | Moderate |
| Huffines 2013[21] | + | + | + | − | + | ? | ? | ? | + | + | + | + | + | + | ? | ? | ? | ? | ? | Moderate |
| Jacobowski 2010[33] | + | ? | + | ? | + | ? | ? | ? | − | + | + | + | + | + | + | + | + | ? | ? | Moderate |
| Marshall 2016[30] | + | + | + | − | ? | − | − | − | − | + | + | + | + | + | + | + | + | ? | + | Moderate |
| Prichard 2015[35] | + | ? | + | ? | + | + | ? | + | + | + | + | + | + | + | + | + | + | + | ? | High |
| Randall-Curtis 2016[26] | + | + | + | + | + | + | + | − | − | + | + | + | + | + | + | + | + | + | + | Very high |
| Shaw 2014[27] | + | ? | + | + | + | ? | ? | ? | − | + | + | + | + | + | ? | + | + | + | ? | High |
| Weber 2018[34] | + | + | + | + | + | ? | ? | ? | − | + | + | + | + | + | ? | ? | ? | ? | ? | High |
| White 2018[31] | + | + | + | + | + | + | + | + | + | + | + | + | + | + | + | + | + | + | + | Very high |

+, low concern; −, high concern; ?, unclear. Quality assessment tool elaborated in Xyrichis et al.[13]

**Table 4** Quality assessment—qualitative studies

| Study id | Aim of evaluation | Sampling | Data collection | Data analysis | Research relations | Findings | Transferability | Relevance and usefulness | Ethics | Quality of information | Quality of study | Overall weighting |
|---|---|---|---|---|---|---|---|---|---|---|---|---|
| Ernecoff[20] | + | + | + | + | – | + | + | + | + | + | + | High |
| Frisman[24] | + | + | + | + | – | + | + | + | + | + | + | High |
| Garrouste-Orgeas[23] | + | + | + | + | – | + | + | + | + | + | + | High |
| Marshall[30] | + | + | + | + | – | + | ? | + | + | ? | ? | Moderate |
| Rippin[17] | + | ? | + | + | ? | + | + | + | – | ? | ? | Low |

*+, low concern; –, high concern; ?, unclear. Quality assessment tool elaborated in Xyrichis et al.[13]

Patient outcomes (table 11): three kinds of patient outcomes were reported in three studies involving multicomponent-based, web-based and discussion-based interventions: patient mortality,[22 31] adverse events[19] and length of stay.[19 31] A statistically significant increase in mortality was reported in two studies; specifically one study found an actual percentage increase in the intervention group of 31% (RR: 4.91, CI: 1.55 to 15.51, $p<0.05$),[22] while the other of 7.5% (RR: 1.27, CI: 1.09 to 1.48, p=0.008).[31] A relative reduction in adverse events per 1000 patient days of 29% was also reported[9] with an actual mean difference of −17 (CI: -6.95 to -27.05, $p<0.05$). Length of stay was examined in two studies with conflicting results: one reported a statistically significant reduction (MD: −0.77, CI: −0.69 to −0.87, $p<0.001$),[31] while another did not find any statistically significant difference.[19]

## Qualitative results
### Features that influence effectiveness
The findings from the qualitative studies (n=5) were analysed thematically to identify features of the interventions that participants perceived to influence effectiveness. These included accessibility, simplicity, supplementarity, contextualisation, interprofessionality, consistency, relationship and confidence development. Synthesised into five overarching categories, these can serve as principles to inform the future design and development of more refined family member involvement interventions in ICU: (a) practicality, (b) development, (c) interaction, (d) reflexivity and (e) bridging. We present a high-level summary of these below, with a more detailed qualitative report to be made available in a forthcoming publication.

Practicality: identified as a theme in three studies,[20 23 24] this included intervention accessibility, simplicity, long-term application and repetition. Practicality points to effective interventions needing to be readily available with clear instructions, which are as close to universally identifiable as possible. This may enable the intervention to be used as many times as necessary, benefiting family members over and beyond an individual encounter.

Development: in four studies[17 20 24 30] developing family members' contextual knowledge, or insight, was particularly valuable for effective intervention delivery. Insight was key to developing shared understanding among all those engaged in the treatment process. Insight should go beyond an understanding of the patient's condition to encompass the diverse perspectives of healthcare professionals and the emotional strain which family members naturally undergo.

Interaction: three studies[17 20 24] noted that interventions, either explicitly or implicitly, should encourage verbal and physical interaction between clinicians and family members. However, interprofessional discussions among ICU clinicians, and specifically between medical and nursing staff, appeared equally important. Interventions that foster quality interaction at all levels appear essential for the family to be coherently integrated into the structure of the ICU.

**Table 5** Communication outcome measures

| Study | Jacobowski *et al* 2010 | Weber *et al* 2018 | Shaw *et al* 2014 | White *et al* 2018 | Allen *et al* 2017 |
|---|---|---|---|---|---|
| Intervention measure | Family rounds | Family rounds | Team training | Multifamily support | Family on rounds |
| FS ICU 24—frequency of nurse communication | % highest score<br>Pre: 64%<br>Post: 57%<br>**Change: –7%, p=0.30**<br>**RR: 0.89, CI: 0.70 to 1.12** | % top scores<br>Control: 62%<br>Intervention: 78%<br>**Change: 16%, p>0.05**<br>**RR: 1.24, CI: 0.98 to 1.58** | Mean score<br>Pre: 79.2<br>Post: 87.18<br>**Change: 7.98, p=0.04\*** | | |
| FS ICU 24—frequency of doctor communication | % highest score<br>Pre: 38%<br>Post: 60%<br>**Change: 22%, p=0.004\***<br>**RR: 1.60, CI: 1.18 to 2.17** | % top scores<br>Control: 43%<br>Intervention: 56%<br>**Change: 13%, p>0.05**<br>**RR: 1.31, CI: 0.90 to 1.89** | Mean score<br>Pre: 67.86<br>Post: 76.69<br>**Change: 8.83, p=0.04\*** | | |
| FS ICU 24—honesty of information | | | Mean score<br>Pre: 77.78<br>Post: 87.08<br>**Change: 9.30, p=0.01\*** | | |
| Quality of communication (QOC) | | | | Mean score<br>Control: 62.7<br>Intervention: 69.1<br>**Change: 6.39, p=0.001\***<br>**CI: 2.57 to 10.20** | |
| Communication improvement | | | | | Pre: n=49 (100%)<br>Post: n=47 (100%)<br>**Change 0%, p=0.68**<br>**RR: 1** |

FS ICU, family satisfaction intensive care unit; MD, mean difference; RR, relative risk.

Reflexivity: while effective interventions bring people together, three studies[17 23 24] noted they should also benefit the individual family member, who has the opportunity for personal growth in the path towards meaningful family member integration. Specific examples of reflexivity in the featured interventions were identified in the confirmatory exchanges between family members and healthcare professionals, as family members were able to explore their own role within the ICU, the relationships which they had independently developed with their clinicians and how they could use this unique knowledge as a form of leverage in their communication with ICU professionals.

Bridging: effective interventions should also provide key and otherwise unavailable pathways to encourage family member integration. In two studies,[20 23] the supplementary capacity, which effective interventions offer was of particular value when consistent or predictable communication could not be guaranteed. While information availability is important, acknowledging the many difficulties which family members themselves face, and offering ways to overcome these, may ultimately enhance integration and experience.

## Proposed typology
Utilising the learning gained from the current review, and the insights of our service users and carers group, we propose a typology of interventions (figure 2) to be considered when promoting family member involvement in ICU. Presented in a matrix, we position types of interventions along a continuum of low to high involvement (environmental unit changes, web-based support, discussion-based support, multicomponent support, participation in rounds and participation in physical care) along with their key characteristics (type of ICU, family time commitment, professional input, cost, opportunities and challenges), level of evidence (low, moderate and high) and impact on key involvement and health outcomes (satisfaction, decision-making, communication, family well-being,

**Table 6** Decision-making outcome measures

| Study | Huang *et al* 2018 | Shaw *et al* 2014 | Weber *et al* 2018 | Jacobowski *et al* 2010 | Huffines *et al* 2013 | Almoosa *et al* 2009 |
|---|---|---|---|---|---|---|
| **Intervention measure** | Primary physician involvement | Team training | Family rounds | Family rounds | Support care algorithm | Cardiopulmonary resuscitation (CPR) discussions |
| **FS ICU 24—decision-making** | Mean (SD) Control: 80.07 (12.76) Intervention: 81.06 (15.1) **Change: 0.99, p=0.16 CI: −3.00 to 4.98** | Mean score Pre: 77.47 Post: 83.32 **Change: 5.85, p=0.05*** | Mean score (SD) Control: 85.1 (16.3) Intervention: 88.6 (14.6) **Change: 3.5, p=0.20 CI: −1.98 to 8.98** | | | |
| Inclusion in decision-making | % completely satisfied Control: 61.4% Intervention: 75.9% **Change: 14.5%, p=0.05* RR: 1.23, CI: 1.03 to 1.49** | | | % highest score Pre: 66% Post: 76% **Change: 10%, p=0.12 RR: 1.15, CI: 0.96 to 1.38** | | |
| **Control over patient care** | % completely satisfied Control: 55.6% Intervention: 73.6% **Change: 18%, p=0.02* RR: 1.31, CI: 1.07 to 1.61** | | | | | |
| Support in decision-making | | | | % highest score Pre: 49% Post: 69% **Change: 20%, p=0.005* RR: 1.39, CI: 1.09 to 1.79** | | |
| **Share in decisions about care planning** | | | | % scoring excellent Pre: 45% Post: 68% **Change: 23%, p=0.009* RR: 1.52, CI: 1.03 to 2.24** | | |
| Time to decision | | | | | | Days mean (SD) Control: 7.4 (3.2) Intervention: 6.6 (6.5) **Change: −0.8 days CI: −1.48 to 3.08** |

FS ICU, family satisfaction intensive care unit; MD, mean difference; RR, relative risk.

family trauma and patient outcomes). We completed an initial face and content validation with our project's service users group consisting of ICU survivors and family members, and advisory board consisting of ICU professionals, but would still caution that the typology remains in need of further empirical validation in different ICU settings.

## DISCUSSION
The diverse nature of the family involvement interventions we identified, the heterogeneity of research designs found and the varying methodological quality of available studies meant there was not a single intervention that stood out as the recommended way forward. Overall, the available evidence suggests that rounds-based interventions can benefit communication outcomes; multicomponent and web-based interventions can increase family satisfaction and, discussion-based interventions can benefit decision-making and family well-being. However, interventions can be of varying cost, require different kinds and levels of input and time commitment

**Table 7** Satisfaction outcome measures

| Study | Huang *et al* 2018 | Shaw *et al* 2014 | Dykes *et al* 2017 | Weber *et al* 2018 | Huffines *et al* 2013 |
|---|---|---|---|---|---|
| **Intervention measure** | **Primary physician involvement** | **Staff teamwork training** | **Web-based engagement** | **Family rounds** | **Supportive care algorithm** |
| FS ICU 24—global score | Mean (SD) Control: 84.91 (12.17) Intervention: 86.4 (11.76) **Change: 1.49 p=0.16 CI: –2.14 to 5.12** | Mean score Pre: 83.21 Post: 85.69 **Change: 2.48 p=0.32** | Mean score Pre: 84.3 (3) Post: 90 (1.9) **Change: 5.7 p<0.05* CI: 2.31 to 9.09** | Mean score (SD) Control: 86.0 (16.0) Intervention: 90.8 (10.7) **Change: 4.8 p=0.20 CI: –0.12 to 9.72** | |
| HCAHPS | | | % top score 9–10 Pre: 71.8% Post: 93.3% **Change 21.5 p<0.05* RR: 1.33, CI: 1.10 to 1.55** | | |
| Support given | | | | % top scores Control: 54% Intervention: 71% **Change: 17% p>0.05 RR: 1.32, CI: 0.99 to 1.75** | % scoring excellent Pre: 60% Post 75% **Change: 15% p=0.14 RR: 1.23, CI: 0.91 to 1.65** |

| Study | **White *et al* 2018** | **Cray 1989** | **Dalal *et al* 2015** | **Jacobowski *et al* 2010** |
|---|---|---|---|---|
| Intervention measure | **Multicomponent family-support programme** | **Family-support programme** | **Patient-centred toolkit** | **Family rounds** |
| Patient Perception of Patient Centeredness (PPPC) | Mean score Control: 1.8 Intervention: 1.7 **Change: –0.15, p=0.006* CI: –0.26 to –0.04** | | | |
| Satisfaction with intervention | | % Satisfied **100% agreed (76/76)** | % Satisfied **72% (13/18)** | |
| Time to ask questions | | | | % highest score Pre: 40% Post: 23% **Change: –17%, p=0.02* RR: 0.57, CI: 0.37 to 0.90** |

FS ICU, family satisfaction intensive care unit; HCAHPS, Hospital Consumer Assessment of Healthcare Providers and Systems; MD, mean difference; PPPC, Patient Perception of Patient Centeredness ; RR, relative risk.

from clinicians and family members, and impact on different kinds of outcomes. Given the current state of the evidence, we discourage proposing a single intervention as the 'gold standard' and instead encourage the use of our typology to inform decisions based on individual ICU teams' context, needs and available resources.

At this stage, we suggest our typology is used as a reflective tool to aid ICU teams make evidence-informed decisions about potential interventions they may wish to consider adopting locally. In the first instance, teams are advised to discuss the kinds of engagement and/or clinical outcomes their unit wants to improve as a priority so they can focus and limit their choice of intervention types.

They could then consider the evidence level, cost and key opportunities and challenges offered by each intervention type to help them make fully informed decisions as well as manage expectations. Finally, they could consider the time investment expected of ICU professionals and of family members for the different intervention types; for example, in geographically remote areas where family visiting is a challenge and staff shortage is significant, the team may favour trialling an intervention with less time commitment.

Through focussing on the different types of available interventions, their associated involvement and clinical outcomes, features perceived to influence effectiveness,

**Table 8**  Anxiety outcome measures

| Study | Garrouste et al 2016 | Prichard and Newcomb 2015 | White et al 2018 | Randall-Curtis et al 2016 |
|---|---|---|---|---|
| Intervention measure | Interprofessional family conference | Hand massage | Multifamily support | Communication facilitator |
| HADS—anxiety | 90 days, median (IQR) Control: 8 [4.5–12] Intervention: 4 [1-9] **Change: –4, p=0.01*** % with anxiety score>8 Control: 52.3% (n=23) Intervention: 33.3% (n=14) **Change: 19%, p=0.08 RR: 0.95, CI: 0.63 to 1.44** | Mean change in score Control: –0.4 Intervention: –3.87 **Change: –3.47, p=0.002* CI: –5.5 to –1.4** | | |
| HADS—global score | | | Mean score Control: 12.0 Intervention: 11.7 **Change: –0.34, p=0.61 CI: –1.67 to 0.99** | |
| GAD 7—anxiety | | | | Mean score 3 months Control: 3.0 Intervention: 2.3 **Change: –0.7, p=0.50 CI: –2.91 to 1.42** Mean score 6 months Control: 2.7 Intervention: 1.8 **Change: –0.9, p=0.43 CI: –3.10 to 1.32** |

GAD-7, Generalised Anxiety Disorder Assessment; HADS, Hospital Anxiety and Depression Score; IP, interprofessional; MD, mean difference; RR, relative risk.

and proposing an overarching typology of family involvement interventions, the current review makes a significant contribution to the available literature. Past review work in this area largely focused on interventions that can support family members of critically ill patients;[36–39] these included providing care and treatment options for family members to counter the negative effects arising out of a loved one being hospitalised, or dying, in ICU. Available guidelines often suggest a long list of initiatives to support family members without full consideration or instruction on the contextual requirements of these suggestions or the kind of impact likely to be achieved by these. While prior reviews concluded, as have we, that the evidence in this field needs development and strengthening, these did not offer a granulated analysis of the evidence level for different kinds of interventions or presented these in

**Table 9**  Depression outcome measures

| Study | Garrouste et al 2016 | Prichard and Newcomb 2015 | Curtis et al 2016 |
|---|---|---|---|
| Intervention measure | Interprofessional family conference | Hand massage | Communication facilitator |
| HADS—depression | At 90 days, median (IQR) Control: 5.5 [1–11.5] Intervention: 2 [0–6] **Change: –3.5, p=0.04*** % sign. Depression score>8 Control: 38.6% (n=17) Intervention: 23.8% (n=10) **Change: 14.8%, p=0.14 RR: 0.61, CI: 0.31 to 1.18** | Mean change in score Control: –0.3 Intervention: –2.5 **Change: –2.2, p=0.10 CI: –0.49 to 4.7** | |
| PHQ-9 depression | | | Mean score 3 months Control: 4.9 Intervention: 3.1 **Change: –1.8, p=0.09 CI: –3.89 to 0.31** Mean score 6 months Control: 4.7 Intervention: 2.4 **Change: –2.3, p=0.01* CI: –4.30 to –0.42** |

HADS, Hospital Anxiety and Depression Score; IP, interprofessional; MD, mean difference; PHQ-9, Patient Health Questionnaire; RR, relative risk.

**Table 10** Trauma outcome measures

| Study | Garrouste et al 2016 | Curtis et al 2016 | White et al 2018 |
|---|---|---|---|
| Intervention measure | Interprofessional family conference | Communication facilitator | Multicomponent family support |
| PDEQ | Median (IQR)<br>Control: 14.5([11–23])<br>Intervention: 13 [0–17]<br>**Change: −1.5, p=0.17** | | |
| IES-R | At 90 days, median (IQR)<br>Control: 24 [12.5–45)<br>Intervention: 21([9–33])<br>**Change: −3, p=0.24** | | |
| IES—PTSD (0–88) | | Mean score<br>Control: 20.3<br>Intervention: 21.2<br>**Change: 0.90, p=0.49**<br>**CI: 1.66 to 3.47** | |
| PCL—PTSD | | | Mean score 3 months<br>Control: 31.6<br>Intervention: 29.8<br>**Change: −1.7, p=0.47**<br>**CI: −6.65 to 3.12**<br>Mean score 6 months<br>Control: 30.6<br>Intervention: 27.1<br>**Change: −3.5, p=0.056**<br>**CI: −7.12 to 0.09** |

IES, Impact of Event Scale; MD, mean difference; PCL, Post-Traumatic Stress Disorder Checklist; PDEQ, Peritraumatic Dissociative Experience Questionnaire; PTSD, Post-Traumatic Stress Disorder.

a visual and user-friendly way to inform future work. Such work has helped spark the current movement around family involvement in ICU but remains somewhat limiting in that it does not adequately distinguish between family *support* and family *involvement*, often conflating the two. Consequently, conclusions of past work lack clarity on what can be done to involve as well as support family members.

We expect our typology to introduce some much-needed clarity in this field of research, as well as inform the development of future interventions and research studies. Importantly, the typology may be used by individual ICUs to inform discussions among professionals, and with family members, about which kinds of family involvement interventions they may wish to adopt, trial and implement in their units.

We add to this field of research by adopting strict methodological standards in the current review, to reduce risk of reviewer bias and strengthen transparency. Many of the available guidelines for family involvement in ICU

**Table 11** Patient outcome measures

| Study | Almoosa et al 2009 | White et al 2018 | Dykes et al 2017 |
|---|---|---|---|
| Intervention measure | Cardiopulmonary resuscitation (CPR) discussions | Multifamily support | Patient engagement |
| Death | Control: 8% (3/39)<br>Intervention: 37% (17/45)<br>**Change: 31%, p<0.05***<br>**RR: 4.91, CI: 1.55 to 15.51** | Control: 28.5% (249/873)<br>Intervention: 36% (197/547)<br>**Change: 7.5%, p=0.008***<br>**RR: 1.27, CI: 1.09 to 1.48** | |
| Length of stay | | Mean days<br>Control: 13.5<br>Intervention: 10.4<br>**Change: -3.1, p<0.001*** | Mean (median) (range)<br>Pre: 4.9 (2) (1–108), n=881<br>Post: 5.0 (2) [1–115], n=904<br>**Change: 0.1(0), p=0.61** |
| Adverse events | | | Pre: 59/1000 patient days<br>Post: 42/1000 patient days<br>**Change: -17, % reduction: 29%**<br>**CI: -6.95 to -27.05, p<0.05*** |

RR, relative risk.

are based on a scoping review methodology, which can be methodologically limiting especially in lacking independent and duplicate screening, data extraction and quality assessment; though with notable exceptions.[40] A further way in which we sought to build on others' work was by examining the full spectrum of involvement interventions, rather than focussing on singular aspects such as decision-making. Moreover, we opted to include service users throughout the review process, rather than limiting their involvement to the final stages of peer checking results. This allowed our service users to contribute throughout the process, see how the review was developing, appreciate the technical side of the review process and of the available evidence and importantly to contribute actively in the development of the final typology.

The involvement of a service users and carers group led to us to identify a key, fundamental flaw of existing family involvement interventions and research: in our assessment, these tend to treat family members as a homogeneous group of individuals, rather than as heterogenous units. Specifically, different family members may have different involvement needs and preferences influenced by their previous experiences, knowledge, mental and physical health and socioeconomic state. In addition, the notion of a family member extends beyond the 'next of kin' or 'personal consultee' to encompass extended family, carers and friends. Moreover, our service users emphasised that the dynamics within family units are

crucial to shaping the experience, benefit and satisfaction of family members with ICU care and their level of involvement.

Current family involvement interventions and initiatives appear to be applied in ICUs following a blanket approach, without adequate consideration of particular family members' needs and preparedness for involvement. This may result in some family members benefiting while others experiencing harm, which goes some way in explaining the wide CIs identified in our review that often cross the line of no difference. Indeed, within the PTSD literature, systematic reviews have long concluded that interventions applied universally to victims of trauma, such as compulsory debriefing, should cease and be replaced with a 'screen and treat' model.[41] We therefore invite researchers and clinicians to consider the aforementioned issues carefully in future initiatives for involving family members in ICU.

The current review should be considered in the context of its limitations. These include lack of a meta-analysis, and research published in languages other than English, and from lower- and middle-income countries.

## CONCLUSION

We examined interventions for family member involvement in ICUs, assessed the quality level of the available evidence, presented key results grouped by kinds of outcomes and interventions, and identified key features stakeholders

| Features / Intervention Types | ICU setting | Family time commitment | Professional input | Opportunities & Challenges | Cost | Evidence level | Engagement outcomes | | | Clinical outcomes | | |
|---|---|---|---|---|---|---|---|---|---|---|---|---|
| | | | | | | | Satisfaction | Decision making | Communication | Wellbeing | Trauma | Patient outcomes |
| **Environmental unit changes** Unit change interventions consisted of complete structural redesign of an existing ICU with the objective being enhanced family member presence | General | Low | Low | + Communication + Interaction - Exposure - Privacy | High | Low | Low impact | Very low impact | Low impact | Very low impact | Very low impact | Very low impact |
| **Web-based support** Web based and electronic interventions were approaches which utilised information and communication technology to facilitate information sharing and asynchronous communication | General/ specialist | Moderate | Moderate | + Knowledge + Efficiency - IT proficiency | Moderate | Moderate | High impact | Low impact | Very low impact | Very low impact | Very low impact | Low impact |
| **Discussion-based support** Discussion based interventions generally consisted of one off or regular face to face conversations which took place in the unit between family members and healthcare professionals | General/ Specialist | Moderate | High | + Support + Wellbeing + Decision making - Training | Moderate | High | Moderate impact | Very high impact | High impact | Moderate impact | Moderate impact | Low impact |
| **Multicomponent support** Multicomponent interventions comprised more than one technique to engage, educate and communicate with family members in the ICU | General/ Specialist | High | High | + Family trauma + Pt outcomes - Complexity - Resources | High | High | Moderate impact | Very low impact | Low impact | Moderate impact | Low impact | Moderate impact |
| **Participation in medical rounds** Family involvement in rounds interventions enabled the family members to attend and watch the daily rounds process, with opportunity to raise questions and clarify issues | General ICU | High | Moderate | + Knowledge + Efficiency - Time - Satisfaction | Low | High | Moderate impact | Very low impact | High impact | Low impact | Very low impact | Very low impact |
| **Participation in physical care** Physical participation in care interventions consisted of physical touch between family members and patients being utilised for reciprocal benefit | General ICU | High | Low | + Satisfaction + Wellbeing - Confidence | Low | Low | High impact | Very low impact | Very low impact | Moderate impact | Very low impact | Very low impact |

**Figure 2** Typology of family involvement interventions.

perceived to influence intervention success. We summarised our results in the form of a typology, which we offer here as a first step towards developing a more systematic programme of research on interventions for family involvement in ICU. Based on the learning gained from the current review, we argue that future interventions should be developed with much closer family member input and designed by considering the key success features we identified. Importantly, we call for future interventions to be multilayered to allow for a greater or lesser level, and different kinds, of involvement for family members; the decision of which should be informed by a baseline diagnostic of family members' readiness and preparedness for involvement.

**Acknowledgements** We are grateful to our Expert Advisory Board and patient advisers, members of our Service Users and Carers Group, for their invaluable insight and support throughout this review. We owe intellectual debt to the late Professor Scott Reeves who generously supervised this project until his untimely passing in May 2018.

**Contributors** AX was responsible for conceiving the review, developing the review question and methodology, leading the funding application and coordinating the project. SF and JP were responsible for developing and executing the search strategy. AX, SF and JP were responsible for reference screening, quality assessment, data extraction and management. AMR, MT and SB gave advice throughout the project and contributed to data interpretation. AX and SF were responsible for the statistical inferences and qualitative synthesis, respectively. SF and AX led the development of the first draft of the manuscript. SB, JP, MT and AMR gave editorial comments on the draft manuscript. AX is the guarantor of the review. AX, SF, JP, SB, MT and AMR read and approved the manuscript.

**Funding** This paper presents independent research funded by the National Institute for Health Research (NIHR) under its Research for Patient Benefit (RfPB) Programme (Grant Reference Number PB-PG-0416–20021).

**Competing interests** None declared.

**Patient consent for publication** Not required.

**Provenance and peer review** Not commissioned; externally peer reviewed.

**Data availability statement** Data are available upon reasonable request. All data relevant to the study are included in the article or uploaded as supplementary information.

**ORCID iD**
Andreas Xyrichis http://orcid.org/0000-0002-2359-4337

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
