## [Reviewer comments · BMJ Open]

ARTICLE DETAILS

TITLE (PROVISIONAL)	Interventions to promote family member involvement in adult critical care settings: A systematic review.
AUTHORS	Xyrichis, Andreas; Fletcher, Simon; Philippou, Julia; Brearley, Sally; Terblanche, Marius; Rafferty, Anne Marie

VERSION 1 – REVIEW

REVIEWER	Madeline Schmitt University of Rochester School of Nursing-University of Rochester Medical Center. USA
REVIEW RETURNED	09-Aug-2020

GENERAL COMMENTS	This systematic review article, focused on interventions to promote family involvement in adult critical care settings, is a careful and useful synthesis of the 21 quantitative and qualitative studies that survived the eligibility screening process. The search was quite comprehensive, starting with 5, 109 citations. It appears limited to literature in English. Of the four objectives set for the review [conducting the search, assessing the empirical evidence, providing a synthesis of the interventions and their outcomes, and creating a typology of interventions to support future research], the last is the least successfully achieved. Depending on the results of the review of studies and feedback from a panel of service users, the authors propose a typology that is intended to bring greater clarity to factors that may influence choice of interventions as well as a variety of outcomes, but the processes by which this typology emerged are poorly described, especially how the feedback from service users was incorporated. However, the authors do mention, at the end of the article, that consultation with service users helped them identify a “fundamental flaw” in the reviewed studies- that is the treatment of “family” as a unitary variable, instead of recognizing that the dynamics among family members and who from within the family is included in the interventions can shape the nature of and impact of family interventions on both patients and family members themselves. This awareness is not incorporated into the typology presented. For this draft of the article there are some specific detail recommendations to improve its presentation: 1. Page 4, second paragraph in Background section- referents for the word “this”?2. Page 5 line 2. I believe ICU should be ICUs3. I am unfamiliar with the insertion of “Review registration” information on Page 6, but that may just be my unfamiliarity with BMJ open format.4. Figure 1 seems incomplete as a flow diagram5. It would be helpful to identify the Table referents in the narrative and number the Tables in the order of discussion in the text.
---

	6. Throughout the narrative the authors anthropomorphize studies, instead of indicating that it was authors/researchers who engaged in various actions. 7. I am not clear why #36 is included in the reference list? I do not find it cited in the text, and it is the only non-English reference.
--	--

REVIEWER	Émilie Gosselin Université de Sherbrooke, Canada
REVIEW RETURNED	14-Aug-2020

GENERAL COMMENTS	General comments This is an interesting systematic review about involvement of family members in adult critical care settings. The search and analysis were done with rigor and reported clearly in the manuscript. Some minor comments should be addressed before publication. Specific comments Abstract Results: It would be interesting to see the intervention identified in the review here, as it is the purpose of the study. Conclusion: The last sentence is long and presents many ideas. Please separate in 2 sentences. Background Overall, the background reads well, is clear and concise. I appreciated the definition of "family member involvement" appearing early in the manuscript. Methods p.5 lines 10-15: "for including family members in the care processes and decisions made in adult critical..." why not stay consistent by using "family involvement"? Study eligibility criteria: It is important that you describe here the inclusion AND exclusion criteria that you used. Looking at figure 1, how did you exclude 3 605 studies based on records? Study eligibility criteria and Figure 1: could you specify what you mean by: "wrong outcomes"; "wrong population"; "wrong setting"? Study eligibility criteria and Figure 1: why was a bundle intervention excluded? You conclude that multi-layered interventions should be developed and tested. It would have been interesting to learn about the effect of a multi layered intervention. Quality assessment: Specify what tools you used for quality assessment. Results Study design: It would be interesting to have an idea of the samples sizes of the studies included. Present it in Table 2 as well. Quality of evidence/Table 4 and 5: These table could use a legend,
--

	for +, - and ?, as well as the tool used for assessing quality of evidence. Table 4 and 5: Studies should be ordered in alphabetical order to be consistent with Table 2 and 3. Quantitative results, p.7, line 57: please reorder the categories in the same order then presented in the text. Involvement outcomes, health outcomes and qualitative results: Overall, I would like to know more in the text which intervention is associated to which outcome. It is your third objective to describe the intervention and their associated outcomes. P.9, line 6: HADS has been used previously. You don't need to repeat the long version of it. Qualitative results: Please add the reference showing which studies contributed to each category throughout the results presentation. I think the first paragraph of discussion is still results, as it is your analysis of the literature. It should be moved in the results section. Further description figure 2 should be presented, as it is the answer to the goal of the study. Discussion The discussion is short. However, normally the discussion should compare results with previous literature and explicitly present the main limitations of the study. Main conclusions and ideas for future research should also be introduced here. For example, what support the statement that family members have different involvement in care? Please revise accordingly. Conclusion The conclusion should summarize the study (goal, main results, main conclusions). For example, the reason for recommending multi-layered intervention should be presented and argued in the discussion section. Please revise accordingly. References The reference list shows recent and relevant papers cited.
--	---

VERSION 1 – AUTHOR RESPONSE

Reviewer 1 comments	Authors' response
The processes by which this typology emerged are poorly described, especially how the feedback from service users was incorporated.	Further text added to clarify this process. Please see additional text in red font .
Page 4, second paragraph in Background section- referents for the word "this"?	'this' has been replaced with 'family member involvement' for greater clarity. Please see change in red font .
Page 5 line 2. I believe ICU should be ICUs	Agreed, and corrected. Change in red font .
I am unfamiliar with the insertion of "Review registration" information on Page 6, but that may just be my unfamiliarity with BMJ open format.	The PROSPERO review registration number is included after the abstract, so this detail has now been removed from the manuscript to avoid any confusion.
Figure 1 seems incomplete as a flow diagram	Figure 1 follows standard PRISMA guidance; no change required.

It would be helpful to identify the Table referents in the narrative and number the Tables in the order of discussion in the text.	Of course. We regret this was not clear in the original submission, for some of the tables. We have now reviewed the Results section and addressed this. Please see revisions in red font .
Throughout the narrative the authors anthropomorphize studies, instead of indicating that it was authors/researchers who engaged in various actions.	That is a fair remark, though presumably the reviewer is referring to personification rather than anthropomorphism. If so, we reviewed the paper and avoided this literary device where possible; however, we retained some instances to avoid going over the word limit. We understand this is not an uncommon approach in such papers published in this journal.
I am not clear why #36 is included in the reference list? I do not find it cited in the text, and it is the only non-English reference.	Thank you for spotting this; removed.
Reviewer 2 comments	Authors' response
Abstract Results: It would be interesting to see the intervention identified in the review here, as it is the purpose of the study.	Intervention categories now added in the abstract. Please see addition in red font .
Conclusion: The last sentence is long and presents many ideas. Please separate in 2 sentences.	Done.
Methods: p.5 lines 10-15: "for including family members in the care processes and decisions made in adult critical..." why not stay consistent by using "family involvement"?	Fair point; changed for consistency. See change in red font .
Study eligibility criteria: It is important that you describe here the inclusion AND exclusion criteria that you used. Looking at figure 1, how did you exclude 3 605 studies based on records? Figure 1: could you specify what you mean by: "wrong outcomes"; "wrong population"; "wrong setting"? Figure 1: why was a bundle intervention excluded?	Additional text has been added in this subsection, for clarification. Please see addition in red font .
Quality assessment: Specify what tools you used for quality assessment.	Please see further detail and supporting references in red font .
Results Study design: It would be interesting to have an idea of the samples sizes of the studies included. Present it in Table 2 as well.	Sample size included in the Table and ranges given in the text. Please see additions in red font .
Quality of evidence/Table 4 and 5: These table could use a legend, for +, - and ?, as well as the tool used for assessing quality of evidence. Table 4 and 5: Studies should be ordered in alphabetical order to be consistent with Table 2 and 3.	Legend and citation for the quality assessment tool included. Studies now shown in alphabetical order. Please see addition to the tables in red font .
Quantitative results, p.7, line 57: please reorder the categories in the same order then presented in the text.	Done.
Involvement outcomes, health outcomes and qualitative results: Overall, I would like to know more in the text which intervention is associated to which outcome. It is your third objective to describe the intervention and their associated outcomes.	Interventions have been identified in the text, with additional tables now included in the Supplement which group and present outcomes by intervention type. Please additions in red font .
P.9, line 6: HADS has been used previously. You	Thank you for spotting this, changed.

don't need to repeat the long version of it.	
Qualitative results: Please add the reference showing which studies contributed to each category throughout the results presentation.	References added, seen in red font.
I think the first paragraph of discussion is still results, as it is your analysis of the literature. It should be moved in the results section.	Agreed, and moved.
Further description figure 2 should be presented, as it is the answer to the goal of the study.	Agreed, and have elaborated on this. Please see additional text in red font.
Discussion	
The discussion is short. However, normally the discussion should compare results with previous literature and explicitly present the main limitations of the study. Main conclusions and ideas for future research should also be introduced here. For example, what support the statement that family members have different involvement in care? Please revise accordingly.	Expanded the Discussion further to include previous literature, suggestions for future research and study limitations. All additions in red font.
Conclusion	
The conclusion should summarize the study (goal, main results, main conclusions). For example, the reason for recommending multi-layered intervention should be presented and argued in the discussion section. Please revise accordingly.	Added a few summary lines as suggested, seen in red font.

VERSION 2 – REVIEW

REVIEWER	Madeline H Schmitt University of Rochester School of Nursing Rochester, NY, USA
REVIEW RETURNED	27-Nov-2020

GENERAL COMMENTS	This is a complex and multi-faceted study. The revision is very responsive to previous reviewer and editor concerns. The additional details greatly contribute to further clarity. I especially appreciate the additional information under the Results section "Proposed typology" and the expanded discussion section. Many good additional points are made but it is now almost too many words. I think some editorial revision could improve flow of content, reduce redundancy, and better place some of the content in the appropriate section [e.g., Results or Discussion]. Here are some suggestions for consideration:" On page 15, bottom, starting with "At this stage....The rest of that paragraph and the next one belong in the Discussion section. Relocate from p. 15-top of 16 beginning with 'At this stage... "to end of that paragraph on p.16 and make it the second paragraph in the Discussion section. Then, relocate the last paragraph to before the Discussion section where it begins "We add..." on p.16. Also, insert content from page 17 beginning with "Available guidelines often suggest...way to inform future work" into P.16 before the line that begins "Such work has helped spark..."Delete the first part of paragraph on P.17 beginning with "Much of the available literature...different kinds of outcomes." [point has already been made and this is redundant]. On p. 17 beginning with "By having an engaged service user....face-to-face discussions" move the inclusive content to p.11 and put at the end of the Methods Section under the label Patient and Public Involvement and eliminate redundancy. The rest of the paragraph on P.17 beginning with [added] ["The
---

	involvement of a service users and carers group] led us to identify a key, fundamental flaw...and their level of involvement." can be inserted later on the page after the sentence starting with "Current family involvement....preparedness for involvement." P. 18 first line : Should it say "should cease and be replaced..." Unless the journal requires it I would drop the Conclusion Section. It does not add anything. Conclusions and recommendations have been integrated into the Discussion section. The overall point of the editorial suggestions is to reorganize the added content with what was originally there for flow of ideas and reduction of redundancy. What is offered here is one set of suggestions for doing so.
--	--

REVIEWER	Émilie Gosselin Université de Sherbrooke, Canada
REVIEW RETURNED	21-Nov-2020

GENERAL COMMENTS	Thank you for addressing my original comments. I am satisfied with the revision suggested.
--

VERSION 2 – AUTHOR RESPONSE

Reviewer 1 comments

1. This is a complex and multi-faceted study. The revision is very responsive to previous reviewer and editor concerns. The additional details greatly contribute to further clarity. I especially appreciate the additional information under the Results section "Proposed typology" and the expanded discussion section.

*Authors' response: Thank you for your generous comments.

2. Many good additional points are made but it is now almost too many words. I think some editorial revision could improve flow of content, reduce redundancy, and better place some of the content in the appropriate section [e.g., Results or Discussion]. Here are some suggestions for consideration:

- On page 15, bottom, starting with "At this stage....The rest of that paragraph and the next one belong in the Discussion section.
- Relocate from p. 15-top of 16 beginning with 'At this stage... "to end of that paragraph on p.16 and make it the second paragraph in the Discussion section.
- Then, relocate the last paragraph to before the Discussion section where it begins "We add..." on p.16.
- Also, insert content from page 17 beginning with "Available guidelines often suggest...way to inform future work" into P.16 before the line that begins "Such work has helped spark..."
- Delete the first part of paragraph on P.17 beginning with "Much of the available literature...different kinds of outcomes." [point has already been made and this is redundant].
- On p. 17 beginning with "By having an engaged service user....face-to-face discussions" move the inclusive content to p.11 and put at the end of the Methods Section under the label Patient and Public Involvement and eliminate redundancy.
- The rest of the paragraph on P.17 beginning with [added] ["The involvement of a service users and carers group] led us to identify a key, fundamental flaw...and their level of involvement." can be inserted later on the page after the sentence starting with "Current family involvement... preparedness for involvement."
- P. 18 first line : Should it say "should cease and be replaced..." We appreciate you taking the time to consider our text in such detail, and for your editorial suggestions.

*Authors' response: You make some excellent points here, which indeed help improve the flow of the

argument and avoid redundancy. We have therefore heeded your advice and shifted the text accordingly. Edits can be seen in the Discussion section of the manuscript, in red font.

Reviewer 2 comments

Thank you for addressing my original comments. I am satisfied with the revision suggested.

*Authors' response: Thank you.